# DNA Methylation-Based Testing in Liquid Biopsies as Detection and Prognostic Biomarkers for the Four Major Cancer Types

**DOI:** 10.3390/cells9030624

**Published:** 2020-03-05

**Authors:** Vera Constâncio, Sandra P. Nunes, Rui Henrique, Carmen Jerónimo

**Affiliations:** 1Cancer Biology & Epigenetics Group—Research Center, Portuguese Oncology Institute of Porto (CI-IPOP), 4200-072 Porto, Portugal; veraconstancio24@gmail.com (V.C.); sandra22nunes@gmail.com (S.P.N.); rmhenrique@icbas.up.pt (R.H.); 2Master in Oncology, Institute of Biomedical Sciences Abel Salazar, University of Porto (ICBAS-UP), 4050-313 Porto, Portugal; 3Department of Pathology, Portuguese Oncology Institute of Porto, 4200-072 Porto, Portugal; 4Department of Pathology and Molecular Immunology, Institute of Biomedical Sciences Abel Salazar–University of Porto (ICBAS-UP), 4050-313 Porto, Portugal

**Keywords:** lung cancer, breast cancer, colorectal cancer, prostate cancer, DNA methylation, biomarker, detection, prognosis, liquid biopsy, cell-free DNA

## Abstract

Lung, breast, colorectal, and prostate cancers are the most incident worldwide. Optimal population-based cancer screening methods remain an unmet need, since cancer detection at early stages increases the prospects of successful and curative treatment, leading to a lower incidence of recurrences. Moreover, the current parameters for cancer patients’ stratification have been associated with divergent outcomes. Therefore, new biomarkers that could aid in cancer detection and prognosis, preferably detected by minimally invasive methods are of major importance. Aberrant DNA methylation is an early event in cancer development and may be detected in circulating cell-free DNA (ccfDNA), constituting a valuable cancer biomarker. Furthermore, DNA methylation is a stable alteration that can be easily and rapidly quantified by methylation-specific PCR methods. Thus, the main goal of this review is to provide an overview of the most important studies that report methylation biomarkers for the detection and prognosis of the four major cancers after a critical analysis of the available literature. DNA methylation-based biomarkers show promise for cancer detection and management, with some studies describing a “PanCancer” detection approach for the simultaneous detection of several cancer types. Nonetheless, DNA methylation biomarkers still lack large-scale validation, precluding implementation in clinical practice.

## 1. Introduction

During the last decades, efforts have scaled up worldwide to develop more effective biomarkers as an approach to reduce cancer mortality [1]. A cancer biomarker can be defined as an objectively measurable biomolecule, such as a protein, metabolite, RNA, DNA, or an epigenetic alteration, found in body fluids or tissues, that indicates the presence of cancer or provides information on cancer’s expected future behavior [2,3].

Tissue biopsy sampling has been the gold-standard approach for patients’ diagnosis and prognostication. However, several drawbacks have been pointed out over the years to this approach [4,5]. Firstly, and mainly, tissue samples might not fully represent tumor heterogeneity, constituting a limitation for accurate outcome prediction and treatment efficacy [5,6]. Moreover, early-stage tumor, residual disease, and early recurrence detection might be difficult, since tissue biopsy sampling requires a highly invasive intervention with risk of complications, and, depending on the tumor anatomical location, may be extremely difficult to obtain [4,7]. Therefore, a minimally invasive method that allows for cancer detection at an early stage and patients’ follow-up is essential. Recently, liquid biopsies obtained from easily assessable body fluids, including blood, urine or sputum, have surfaced as a viable alternative to overcome these challenges **[8]**.

Liquid biopsies, mainly based on circulating cell-free DNA (ccfDNA), circulating tumor cells (CTCs), circulating cell-free RNA (ccfRNA) and exosomes [8] are a fast, reliable, cost-effective and minimally invasive approach [9] (Figure 1). Hence, owing to these features, they may allow for a real time monitoring of the cancer evolution, while better representing the heterogeneous genetic profile of all tumor sub clones [8,10]. Furthermore, it has been shown that these molecules contain alterations present in the tumor itself, including mutations [11] and epigenetic alterations [12]. Indeed, due to its early onset, cancer specificity, biological stability, and accessibility in bodily fluids, aberrant DNA methylation has called attention as epigenetic-based biomarker, making it an attractive target to be studied in liquid biopsies [3].

Lung (LC), breast (BrC), colorectal (CRC) and prostate (PCa) cancers are the most incident and among the deadliest worldwide, despite the efforts for early cancer detection and the emergence of new therapies [13]. Given the shortcomings of current screening methods and prognostic biomarkers, the development and implementation of effective biomarkers for cancer detection at curative, early stages, and for better patients’ stratification, is crucial. Thus, for the purposes of this review, a detailed and extensive literature review was conducted aimed to explore the state of the art of ccfDNA methylation blood-based biomarkers for cancer screening, diagnosis, prognosis, prediction, and monitoring of the four most incident cancers worldwide. On a Pubmed database search, the key words “Lung Cancer/Breast Cancer/Colorectal Cancer/Prostate Cancer”, “DNA methylation”, “Diagnosis/Detection/Prognosis” and “Serum/Plasma” were used (Figure 2).

## 2. Circulating Cell-Free DNA Liquid Biopsies

CcfDNA was firstly described in 1948 when extracellular nucleic acids were found in human blood from healthy individuals by Mandel and Métais [14]. Later on, it was found that, in cancer patients, circulating tumor DNA (ctDNA) fragments between 150 and 1000 base pairs could also be detected due to their release into the bloodstream, either by cell death (apoptosis or necrosis) or active secretion by the release of extracellular vesicles, such as exosomes [10,15]. Depending on several factors including tumor burden, metastatic sites or cellular turnover, ctDNA might account for 0.01% to 90% of the total ccfDNA in the blood of cancer patients [15]. Owing to the fact that ctDNA might represent tumor-specific genetic and epigenetic alterations of all tumor’s sub clones present, ccfDNA is an ideal candidate for blood-based liquid biopsies by offering the possibility to test for the presence of cancer and the discrimination of lethally aggressive cancer [16].

## 3. DNA Methylation

Although the study of tumor mutations was the focus of biomarker research for a long time, their wide diversity has been a challenge for the development of effective diagnostic biomarkers since very large proportions of the genome would need to be examined in order to provide adequate sensitivity [17]. Contrarily, epigenetic alterations seem to be more stable and homogenous in cancer, representing a good alternative for biomarker development [18]. The main studied epigenetic mechanisms are DNA methylation, histone post-translational modifications, histone variants and chromatin remodeling complexes (Figure 3) [19]. Although these mechanisms are crucial for normal cell development and regulation of specific gene expression patterns, epigenetic dysregulation often leads to inappropriate activation or inhibition of several signaling pathways, which may trigger the development of several pathologies, including cancer [20].

DNA methylation, the most widely studied epigenetic modification in humans, was also the first to be identified in cancer [20,21]. This epigenetic mechanism consists of covalent addition of a methyl group, donated by S-adenosylmethionine (SAM), to the 5-position carbon of a cytosine ring to form 5-methylcytosine (5mC) [22,23]. This modification is catalyzed by DNA methyltransferase enzymes (DNMTs), namely, DNMT3a and DNMT3b that catalyze *de novo* DNA methylation during embryonic development, establishing tissue-specific DNA methylation, and DNMT1 that is often associated with maintenance of methylation patterns during replication [23]. Typically, this process occurs on cytosine residues present at CpG dinucleotides commonly found in large clusters named CpG islands, which are predominantly located at the 5′ end of genes, occupying approximately 60% of human gene promoter regions [20,23,24]. Although gene promoter hypermethylation is associated with transcription repression of the nearby gene (Figure 4), depending on DNA methylation genomic location, it can display different functions [22,25]. Epigenetic gene silencing by DNA promoter methylation may occur either directly, by blocking transcription factors to prevent binding to target sites in or near the promoter, or indirectly, through binding of methyl-CpG-binding proteins (MBP), which can recruit other enzymes like DNMTs and histone deacetylases (HDAC), leading to chromatin conformation changes that further repress gene transcription [20,22].

As aforementioned, DNA methylation is crucial for multiple cellular processes, thus it is understandable that its deregulation has been linked to cancer. Indeed, normal and cancer cells display different methylomes. Usually, a global hypomethylation pattern, which contributes to genomic instability and activation of silenced oncogenes, is observed in cancer [23,26]. Alongside, tumor suppressor genes (TGS) frequently undergo inactivation due to focal promoter hypermethylation [23,26]. Currently, the latter process is considered a major contributor to neoplastic transformation [27].

Interestingly, aberrant DNA methylation is thought to occur at very early stages of cancer development and specific genes seem to be methylated at different tumor stages [23]. Moreover, since these alterations can be assessed in several body fluid samples [23], it is widely accepted that DNA methylation-based liquid biopsies are a promising approach, not only for premalignant/early cancer detection, but also for prognostic assessment. Furthermore, since some genes seem to acquire tissue-specific DNA methylation, it may also be possible to discriminate between different cancer types in the context of metastatic tumors [23] or in liquid biopsies.

## 4. Cell-Free DNA Methylation-Based Biomarkers

### 4.1. Lung Cancer

#### 4.1.1. Screening and Diagnosis

Despite advances in new treatment options over the years, the high mortality rate observed in LC patients is mainly related to the fact that more than 75% of LC patients are diagnosed with advanced stage disease [28]. Hence, effective screening options aiming to shift LC diagnosis from advanced to curative early stages are crucial to change the fate of this disease [29]. Currently, low-dose computed tomography (LD-CT) is considered the best LC screening method available [30]. Nonetheless, despite The National Lung Screening Trial has shown a 20% decrease in LC-related mortality rate among high-risk smokers with LD-CT screening comparing to chest X-ray [corroborated by the largest European trial (Dutch-Belgian Lung Cancer Screening Trial)] from the 24% positive test results in this trial, 96.4% were deemed as false-positive [31,32]. Hence, due to the risks related to LD-CT screening, namely, overdiagnosis, radiation exposure, and false positive results leading to unnecessary anxiety and costs [29], the development of specific and accurate screening tools is urgently needed to improve LC survival.

In 2002, Usadel et al. and Bearzatto et al. reported, for the first time, *APC_me_* and *p16^INK4a^_me_*, respectively, in ccfDNA as putative minimally-invasive biomarkers for LC detection [33,34]. Thenceforth, numerous other methylated gene promoters detected in ccfDNA have been proposed for LC detection either individually or in panel (Table 1). *RASSF1A_me_* and *p16^INK4a^_me_* represent the two most frequently reported genes in blood-based liquid biopsies displaying 22–66% sensitivity and 57–100% specificity for LC detection, individually [34,35,36].

After the commercialization in Europe of a test based on *SHOX2_me_* assessment in bronchial aspirates, the methylation of this gene was also evaluated as an LC detection biomarker in plasma. Indeed, plasma *SHOX2_me_* discriminated LC from control samples with 60% sensitivity and 90% specificity, although higher sensitivity was found in stages II (72%), III (55%) and IV (83%) compared with stage I (27%) patients [37]. In line with these results, another study reported that *SHOX2_me_* discriminated LC in subjects undergoing bronchoscopy with 81% sensitivity and 79% specificity [38]. Later on, Weiss et al. reported that *SHOX2_me_* and *PTGER4_me_* panel distinguished LC patients with 67% sensitivity for a fixed specificity of 90%, and with 73% specificity for a fixed sensitivity of 90% [39]. Remarkably, in the end of 2017, the “Epi proLung^®^” assay, developed by Epigenomics AG, based on these two genes received the *Conformité Européenne* (CE) mark for In Vitro Diagnostic (IVD) test. According to their validation study comprising 360 patients from the US and Europe, of which 152 were diagnosed with LC, depending on the Epi proLung^®^ test score threshold chosen, 85% sensitivity was achieved for 50% specificity, whereas sensitivity decreased to 59% if 95% specificity was considered [40,41].

Although the majority of these studies have focused on detection of non-small cell lung carcinoma (NSCLC) (Table 1) (the most diagnosed LC subtype), different detection frequencies of genes’ methylation have been reported among the different subtypes. Indeed, *SHOX2_me_* detected with higher sensitivity small cell lung cancer (SCLC) (80%) and lung squamous cell carcinoma (LUSC) (63%) than lung adenocarcinoma (LUAD) (39%) [37]. Similarly, *DCLK1_me_* was more frequent in SCLC than NSCLC [42], and our research team recently reported higher *APC_me_* and *RARβ2_me_* levels in SCLC compared to LUAD, in females [36]. Conversely, *SEPT9_me_* was more frequent in NSCLC (53%) than in SCLC (26%) [43]. A serum-based gene panel (*MARCH11_me_, HOXA9_me_, CDO1_me_, UNCX_me_, PTGDR_me_* and *AJAP1_me_*) detected stage I LUAD and LUSC with 71% specificity and 72% and 60% sensitivity, respectively [44]. Interestingly, *HOXA9_me_*, and *RASSF1A_me_* were able to discriminate SCLC from NSCLC with 64% and 52% sensitivity and 84% and 96% specificity respectively [45], whereas *HOXD3_me_* and *RASSF1A_me_* panel reached 75% sensitivity and 88% specificity in male samples [46].

#### 4.1.2. Prognosis, Prediction, and Monitoring

Besides LC histological subtype, TNM prognostic stage groups remain the most important prognostic feature to predict recurrence and survival, followed by tumor histological grade, gender and age [63,64]. Recently, molecular subtypes have also emerged to provide more personalized genetic information, which may improve prognostic estimates and therapy response prediction [63].

Thus far, only a few small-scaled studies reported the prognostic, predictive and monitoring potential of ccfDNA methylation for LC, most of them performed in patients with advanced disease. *DCLK1_me_* and *SOX17_me_* levels were associated with reduced overall survival (OS) in advanced LC and NSCLC, respectively [42,59]. Likewise, higher *SHP1P2_me_* levels in advanced NSCLC associated with reduced progression-free survival (PFS) and OS [65], whereas *BRMS1_me_* associated with both reduced disease-free survival (DFS) and OS in operable NSCLC, and reduced PFS and OS in advanced NSCLC [66].

Interestingly, after neoadjuvant chemotherapy and surgery with intraoperative radiation therapy, NSCLC patients showed decreased *RASSF1A_me_* and *RARβ2_me_* levels, similar to levels in healthy subjects. Moreover, methylation levels increase in at least one of these genes, up to the levels detected before treatment, was observed in all five patients, which disclosed evidence of disease progression [35]. Increased *APC_me_* and/or *RASSF1A_me_* levels within 24h after cisplatin-based chemotherapy also associated with increased OS [67]. Detectable circulating levels of *APC_me_* and *RASSF1A_me_* panel at diagnosis was also found to be an independent predictor of increased disease-specific mortality in LC patients, displaying a 3.9-fold risk of dying from LC comparing to those lacking methylation [46].

Remarkably, advanced LC patients that clinically responded to chemo/radiotherapy demonstrated a decrease in *SHOX2_me_* plasma levels, observed at 7–10 days after therapy initiation [68]. Additionally, higher *SHOX2_me_* levels, both before and 7–10 days after therapy beginning were indicative of shorter OS [68]. Similar results were obtained after two cycles of chemotherapy or TKI-based targeted therapy, being *SHOX2_me_* levels before therapy again predictive of OS [69]. Contrarily, *14-3-3σ_me_* levels in stage IV NSCLC patients before treatment with cisplatin-gemcitabine associated with longer survival [70]. Stage IV NSCLC patients with unmethylated *CHFR* depicted longer OS when treated with EGFR-TKI compared to those treated with chemotherapy, as second-line therapy [71].

### 4.2. Breast Cancer

#### 4.2.1. Screening and Diagnosis

BrC is estimated to be the most incident and deadly cancer in females, worldwide [13]. BrC incidence increased after the implementation of mammography-based screening [72]. Screening mammography endures a sensitivity around 82% and 91% specificity [73]; however, sensitivity decreases with high breast density [74]. Furthermore, although BrC screening features several benefits, it presents some important disadvantages, including overdiagnosis and false-positive results. Indeed, about 11% of cases detected in a population invited to screening would probably not be clinically relevant in the woman’s lifetime, still are treated [75]. Additionally, false-positive results lead to extra imaging exams and eventually biopsy procedures, which can cause discomfort and anxiety to the subjects [75,76]. The diagnosis of BrC comprises clinical examination and biopsy procedures either by image-guided core needle biopsy or fine-needle aspiration, to confirm the diagnosis by histopathological analysis [72]. Since these are invasive procedures, advances in BrC detection are needed, specifically in pre-screening methods, which might select patients to invasive/costly screening tests, avoiding overdiagnosis and unnecessary exams.

Several methylated genes have been proposed as tumor biomarkers for BrC detection [77,78,79,80]. Nevertheless, the sensitivity for cancer detection of one methylated gene in ccfDNA is limited, hence several studies attempted to assemble gene methylation panels to increase the test sensitivity [77,78,79,80]. The currently reported ccfDNA methylation biomarkers for BrC are displayed on Table 2. One of the first studies reported 62% sensitivity and 87% specificity for BrC detection in plasma samples by assessing *APC_me_*, *GSTP1_me_*, *RARβ2_me_* and *RASSF1A_me_* [77]. Nevertheless, a panel including *DAPK1_me_* and *RASSF1A_me_* showed the highest sensitivity, *i.e.*, 96% for BrC detection using methylation-specific PCR (MSP) [81]. Other studies attempted to assemble panels for BrC detection using quantitative methylation-specific PCR (qMSP), a quantitative method, achieving sensitivities above 80% [78,82]. Furthermore, several panels reached 100% specificity for BrC detection [83,84,85]. Altogether, *APC_me_*, *RARβ2_me_* and *RASSF1A_me_* are the most reported genes for BrC detection [77,83,86].

Interestingly, Shan et al. reported a six-gene panel that detects BrC with a tumor size ≤ 1cm with higher sensitivity (85.82%) than mammography (81.82%) [87]. Even though numerous studies report the feasibility of DNA methylation to detect BrC, its validation and transfer to the clinical setting is still overdue. Nonetheless, DNA methylation shows promise to complement BrC standard screening and diagnosis methods.

#### 4.2.2. Prognosis, Prediction, and Monitoring

In addition to staging, several biomarkers are used to predict BrC patients’ response to therapy and prognosis, including histological grade—a high histological grade is associated with poorly differentiated tumors and worse prognosis [96]. Furthermore, 75% of BrC cases are estrogen receptor (ER)-positive, being less aggressive, and associated with a better prognosis [97]. Accordingly, progesterone receptor (PR)-positive tumors are frequently ER-positive. On the contrary, ER-negative and PR-negative tumors are frequently grade 3, often recur, and do not respond to hormonotherapy [96]. Erb-B2 receptor tyrosine kinase 2 (ERBB2) [or human epidermal growth factor receptor 2 (HER2)] is a growth factor overexpressed in 15% of all BrC cases [97]. Hence, therapies targeting ERBB2 such as trastuzumab are currently available for BrC patients, improving patient outcome [97]. Moreover, gene expression profiles have emerged to assist in adjuvant treatment decision [64] Oncotype DX^®^, MammaPrint^®^ and PAM50 (Prosigna) are examples of gene expression profiles that allow for BrC classification and prognostic stratification [64]. Nonetheless, their usefulness in clinical practice is limited and therefore their implementation remains restricted [98]. Thus, new and reliable biomarkers to aid in defining a BrC patient prognosis is of major importance.

Chimonidou et al. analyzed *CST6_me_* in ccfDNA from plasma samples and, during follow-up time, *CST6* was methylated in 13 of the 25 BrC patients that relapsed and in 3 of the 9 patients which died, however not reaching statistical significance [99]. Furthermore, *RASSF1A_me_* and *PITX2_me_* were found to be independent biomarkers of poor OS and *RASSF1A_me_* together with lymph node and ER status indicated poor distant DFS in BrC patients’ ccfDNA [100]. Recently, a study analyzing methylation patterns of several genes in ccfDNA showed that BrC patients with positive *SOX17_me_* and *WNT5A_me_* displayed shorter OS, whereas *KLK10_me_* associated with higher number of relapses and shorter disease-free interval [101]. Apart from being a detection biomarker, *SOX17_me_* associated with lymph node metastasis, poor DFS and OS in plasma samples from BrC’s patients [102]. Visvanathan et al. described a cumulative methylation index (CMI) comprising the methylation of 6 genes (*AKR1B1_me_*, *HOXB4_me_*, *RASGRF2_me_*, *RASSF1_me_*, *HIT1H3C_me_*, and *TM6SF1_me_*) that associates with PFS and OS, *i.e.*, PFS and OS were significantly shorter in metastatic BrC patients with high CMI [103]. Additionally, *14-3-3-σ_me_* methylation was significantly associated with the response to metastatic BrC treatment [104]. *APC_me_* and *RASSF1A_me_* were also associated with poor outcome, with a relative risk of death of 5.7 [105].

### 4.3. Colorectal Cancer

#### 4.3.1. Screening and Diagnosis

Due to its slow progression time and the opportunity to easily remove precancerous and early stage cancerous lesions, if caught on time, CRC entail minimal risk to the patient with about 90% long-term survival [106,107]. Therefore, CRC screening programs are of great interest. Currently, analysis of trace blood in stool by fecal occult blood test (FOBT)/fecal immunochemical test (FIT), and internal imaging of the colon by colonoscopy are the main screening options available for CRC early detection [106,108]. Additionally, biopsy samples during colonoscopy are mandatory for histological diagnosis [109]. Nevertheless, fecal screening tests have limited sensitivity to detect precancerous lesions, whereas colonoscopy is very precise and can be used to remove the lesion during the examination, although it is a costly and invasive procedure with low patient compliance [110]. Hence, despite being recognized that screening reduces CRC incidence and mortality, the availability and compliance to the current screening tests remain suboptimal [111].

The increasing knowledge of the influence of epigenetic alterations in malignant transformation in the gut gave rise to an opportunity for development of sensitive and specific minimally invasive epigenetic-based biomarkers for CRC. Hence, plentiful studies have investigated the detection value of these biomarkers (Table 3). *SEPT9_me_* is the mostly reported methylated gene in blood from CRC patients. Remarkably, this marker was the first blood-based IVD assay for detection of occult cancer based on an epigenetic alteration, approved by the US Food and Drug Administration (FDA) in 2016, under the designation “Epi ProColon^®^ 2.0” (Epigenomics AG) [40]. Additionally, this CE-IVD marked test is also commercially available in Europe and China [112]. A meta-analysis published in 2017 reported that *SEPT9_me_* sensitivity for CRC detection varies between 73–78% depending on the algorithm used to consider a positive result, while specificity varies between 84–96% [113]. Nevertheless, when the biomarker performance of this gene’s methylation was assessed in a multicenter screening setting (PRESEPT clinical trial) with asymptomatic individuals older than 50 years old, the results from 53 CRC cases and 1457 subjects without CRC yielded 48% sensitivity and 92% specificity [114].

Given the importance of detecting pre-malignant conditions, several studies have not only studied the CRC detection performance, but also the ability to detect adenomas. Disappointingly, *SEPT9_me_* performance to detect advanced adenomas ranged between 8–31% [115,116,117], being reported to be 11% in the previously mentioned screening setting study [114]. Thus, the usefulness of this gene for population-based screening is questionable.

As expected, gene panels have improved the performance to detect both adenomas and CRC. Remarkably, *APC_me_, MGMT_me_, RASSF2A_me_* and *WIF1_me_* panel discriminated adenomas and early stage CRC with 75% and 87% sensitivity, respectively, and 92% specificity [118], whereas *SFRP1_me_, SFRP2_me_, SDC2_me_*, and *PRIMA1_me_* panel detected adenomas with 89% sensitivity and 87% specificity, and CRC with 92% sensitivity and 97% specificity [119]. Nevertheless, the performance of these panels in a large screening setting remains to be elucidated. Interestingly, *BCAT1_me_* and *IKZF1_me_* panel performance has been evaluated in large multicenter studies, displaying 62–66% sensitivity and 92–95% specificity for CRC detection, although with a limited 6–9% sensitivity for adenoma detection [120,121].

#### 4.3.2. Prognosis, Prediction, and Monitoring

TMN prognostic stage groups, in addition to cancer location, patient’s age and comorbidities are the basis for CRC prognostication and patient management [64,109]. Regarding ccfDNA methylation-based biomarkers, higher *RUNX3_me_* levels associated with lymphatic invasion, advanced pathological stage and tumor recurrence [151]. *TFPI2_me_* and *SFRP2_me_* levels associated with poorly differentiated carcinoma, deep invasion, lymph node and distant metastasis [131,152], the former also associating with tumor size [131], and the latter with shorter OS [152]. *SST_me_* was independently associated with higher recurrence risk and shorter DSS in patients who underwent curative surgical resection [153].

A small-scale study suggested that *p16^INK4a^_me_* could reflect the recurrence status during follow-up after surgery [154]. Interestingly, in a prospective cohort study including 150 stage I-III CRC patients from whom serum samples were obtained 1 week before, as well as 6 months and 1 year after surgery, high levels of *TAC1_me_* after 6 months and *SEPT9_me_* after 1 year were independent predictors of tumor recurrence and shorter DSS [155]. Additionally, the increment of *TAC1_me_* and *SEPT9_me_* levels independently predicted disease recurrence, whereas *NELL1_me_* at both 6 months and 1 year associated with disease-specific survival (DSS) [155]. In the same line, *SEPT9_me_* was suggested as follow-up marker for recurrence and metastasis detection, also associating with tumor size, histological grade and histological type [117]. *GATA5_me_, SFRP2_me_*, *ITGA4_me_*, *SHOX2_me_* and *SEPT9_me_* levels were also associated with tumors’ histological grade, TNM stage and lymph node metastasis [141,156], whereas *GATA5_m_*_e_ also associated with large tumor size [141]. Furthermore, *APC_me,_ SEPT9_me_*, *SHOX2_me_* and *SOX17_me_* levels were also shown to be significantly higher in female patients with metastatic disease [36], whereas higher *RARB2_me_*, *SEPT9_me_* and *SOX17_me_* levels were disclosed in metastatic CRC male patients [46].

Interestingly, various studies have demonstrated the prognostic value of *HLTF_me_* and *HPP1_me_* levels in CRC patients. Indeed, serum *HLTF_me_*, *HPP1_me_* and *hMLH1_me_* were significantly correlated with tumor size, and the two former genes (*HLTF_me_* and *HPP1_me_*) further associated with metastatic disease and tumor stage, as well as with worst outcome [125]. Later, Philipp et al., in a study involving 311 serum samples of CRC patients, reported that *HLTF_me_* and *HPP1_me_* associated with tumor size, stage, grade and metastatic disease, and *HPP1_me_* also associated with nodal status. Moreover, in stage IV patients, high levels of both these genes associated with reduced OS [157]. In patients curatively resected for CRC, pre-therapeutic serum *HLTF_me_* levels were associated with increased relative risk of disease recurrence [158]. In a clinical trial including 467 metastatic CRC patients treated with a combination therapy containing fluoropyrimidine, oxaliplatin and bevacizumab, patients with detectable plasmatic *HPP1_me_* before the start of treatment showed a significantly poorer OS [159]. Moreover, patients in which it reduced to undetectable levels 2–3 weeks after treatment, showed a better OS compared to patients that maintained detectable plasma *HPP1_me_* levels [159], suggesting the usefulness of *HPP1_me_* as a prognostic and early response biomarker. Recently, Barault et al. also suggested that plasmatic methylation changes of *EYA4_me_, GRIA4_me_, ITGA4_me_, MAP3K14-AS1_me_* and *MSC_me_* panel over time correlate with tumor response in metastatic CRC patients treated with chemo- or targeted therapy [160]. Interestingly, using the same panel, Amatu et al. reported that circulating methylated DNA normalized to the total amount of circulating DNA dynamics can reflect clinical response to treatment with regorafenib [161].

Since 2017, COLVERA™, a Laboratory Developed Test (LDT), based on a two-gene panel (*IKZF1_me_* and *BCAT1_me_*) is commercialized in the USA, for post-surgery residual disease detection and CRC patient surveillance [162]. The detection of these two genes in blood associated with stage [120] and showed a rapid reversion after surgical resection (35 of 47 positive patients at diagnosis, became negative after surgery) [163]. Moreover, in patients undergoing surveillance after primary CRC treatment, this panel was positive in 68% plasma samples of the 28 patients with clinically detectable recurrent CRC, whereas CEA was positive in only 32%, although specificity was similar in both tests (87% and 94%, respectively) [164]. Hence, this panel doubled the sensitivity of the current gold-standard marker for CRC monitoring. Recently, positivity of this panel after surgery also independently associated with increased risk of recurrence [165].

### 4.4. Prostate Cancer

#### 4.4.1. Screening and Diagnosis

Curable PCa is mostly asymptomatic, and, thus, patients with clinically detected disease are mostly at advanced stage, resulting in worse outcome and limited treatment options [166]. Digital rectal examination (DRE), in combination with serum prostate-specific antigen (PSA) quantification, remains the gold standard PCa screening tools [167], however both methods present drawbacks. A DRE positive result is dependent on clinicians’ expertise and the majority of cancers detected by this method are at advanced stage [168], besides compliance is rather low. The widespread adoption of PSA screening since the late 1980s has facilitated the shift to detection of PCa at early stages [168]. Nonetheless, despite being highly sensitive, since benign prostatic hyperplasia (BPH) and other benign conditions also cause PSA elevation, the lack of cancer-specificity of this approach entails a high false-positive rate and overdiagnosis of non-life threatening PCa [167,169]. Indeed, only less than one third of the patients undergoing transrectal ultrasound-guided (TRUS) biopsy (standard diagnostic approach) due to elevated PSA levels or abnormal DRE are diagnosed with cancer [170]. In parallel, a negative result does not completely rule out the existence of cancer, leading to a large number of unnecessary invasive tissue biopsies that might be repeated due to the uncertainty of diagnosis if elevated PSA levels persist [170]. Thus, the introduction of more specific alternatives is urgently sought.

Besides blood-based liquid biopsies, aberrant DNA methylation in urological cancers can also be detected in urine, which is a non-invasive, easily accessible source of exfoliated cells and ccfDNA from diverse sites of the urinary system [171]. Hence, several studies have been carried out using this source [171]. Nonetheless, in PCa, higher sensitivities are achieved after manipulation of the prostate, either by prostate massage or DRE, increasing the invasiveness of this procedure [171]. Therefore, blood-based liquid biopsies might represent the better minimally invasive procedure for PCa detection. A summary of the currently reported ccfDNA methylation PCa detection biomarkers is depicted in Table 4.

*GSTP1_me_* is the most frequently described epigenetic alteration in ccfDNA of PCa patients due to its remarkably high specificity for PCa [172,173,174,175,176]. Indeed, a meta-analysis evaluating *GSTP1*_me_ PCa detection performance in plasma/serum reported a pooled specificity of 90% (non-qMSP) and 96% (qMSP-based detection), although with a modest sensitivity of 40% (non-qMSP) and 36% (qMSP-based detection) [174].

Since epigenetic alterations are usually multiple and not necessarily overlapped, multigene panels are pivotal to increase the modest sensitivities of individual genes. Indeed, Ellinger et al. reported that using a gene panel comprising *GSTP1_me_, PTGS2_me_, RPRM_me_* and *TIG1_me_*, PCa diagnostic coverage increased from 42% (*GSPT1_me_* alone) to 47% (panel), maintaining 93% specificity [172]. Furthermore, Sunami et al. reported that *GSTP1_me_*, *RASSF1A_me_* and *RARβ2_me_* were hypermethylated in 13%, 24% and 12% of serum samples from PCa patients, respectively, whereas the three gene panel increased the detection rate to 29%, with 100% specificity [173]. Remarkably, the addition of *FOXA1_me_* to the previous panel increased sensitivity to 72% although at the expense of lower specificity (72%) [46]. More recently, other panels without comprising *GSTP1_me_* have also been tested. Indeed, *MCAM_me_, ERα_me_* and *ERβ_me_* panel disclosed 75% sensitivity and 70% specificity for early PCa detection [177]. Likewise, *ZNF660_me_, CCDC181_me_, ST6GALNAC3_me_* and *HAPLN3_me_* in serum displayed 22%, 26%, 31% and 44% sensitivity, respectively, and 100% specificity for PCa. Remarkably, the best gene panel (*ST6GALNAC3_me_, CCDC181_me_* and *HAPLN3_me_*) increased sensitivity to 67%, keeping 100% specificity [178].

Interestingly, given the modest sensitivities obtained with ccfDNA methylation even with gene panels, studies have also attempted to understand whether ccfDNA might have a better performance by complementing it with serum PSA levels. Indeed, in a Mexican cohort with biopsy-confirmed PCa, a panel comprising *GSTP1_me_* and *RASSF1A_me_* allowed for cancer detection with 73% positive predictive value (PPV) and 59.6% negative predictive value (NPV), increasing to 81% and 66%, respectively, when serum PSA was also considered [179]. Likewise, serum *GADD45a_me_* increased sensitivity from 38% to 94% when PSA and free circulating DNA levels were also considered, although specificity decreased from 98% to 88% [180].

#### 4.4.2. Prognosis, Predicition, and Monitoring

PCa is a very heterogeneous disease, ranging from small, low grade, clinically indolent tumors to large, lethally aggressive ones [184]. Thus, the main goal after establishing the presence of cancer is to evaluate its extension and aggressiveness through grading and staging, to assess prognosis and plan the treatment strategy. Currently, PCa prognostic stage groups are based on nomograms combining TNM classification, preoperative serum PSA levels, and histological International Society of Urological Pathology (ISUP) Grade Group [185,186], which is based on the evaluation of the two most common differentiation patterns in a tumor [187]. Furthermore, it is estimated that 30-50% patients treated with curative intent may show rising serum PSA levels (biochemical recurrence) within 10 years after treatment, with clinical disease progression occurring in up to 40% of these [188,189]. Thus, considering these uncertainties, there is an urgent need for development and implementation of more accurate PCa biomarkers to assist clinicians and patients in decision-making.

Besides the study of its detection value, the potential of methylation-based biomarkers in ccfDNA to predict disease progression and therapy response have also been tackled. *GSTP1_me_* [181,190], *MDR1_me_*, *EDNRB_me_* and *RARβ2_me_* [181] were reported to be more frequent in castration-resistant prostate cancer (CRPC) patients than in early-stage PCa patients. *GSTP1_me_* levels also associated with Gleason score and presence of metastasis [190] as well as with reduced DSS along with *APC_me_* [191] in CRPC patients. Furthermore, *GSTP1_me_* and *RASSF2A_me_* were more frequently detected in men with non-organ confined compared to organ-confined disease and both associated with increased Gleason score [175]. Interestingly, preoperative serum *GSTP1_me_* was also reported as an independent predictor of biochemical recurrence following radical prostatectomy [192]. Sunami et al. reported that *GSTP1_me_, RASSF1A_me_* and *RARβ2_me_* associated with Gleason score and serum PSA levels, whereas *GSTP1_me_* and *RARβ2_me_* also associated with advanced stages of disease [173]. Moreover, individually, serum *PCDH17_me_, PCDC10_me_* and *PCDH8_me_* were also associated with advanced clinical stage, higher preoperative serum PSA, as well as lymph node metastasis and shorter biochemical recurrence-free survival [193,194,195]. Besides being also associated with Gleason score, advanced tumor stage and high PSA, *CDH13_me_* was further associated with shorter OS [183].

Remarkably, a recent phase III multicenter trial enrolling 600 CRPC patients showed that detectable serum *GSTP1_me_* levels prior and after two cycles of chemotherapy were both independently associated with decreased OS [196]. In the same study, undetectable serum *GSTP1_me_* after two cycles of docetaxel further associated with longer time to PSA progression [196].

## 5. Cell-Free DNA Methylation as a Candidate “PanCancer” Screening Biomarker

The screening of these four cancer types faces different challenges. On the one hand, LC and CRC demand an early cancer detection that is still not fully accomplished with the current screening tools available. On the other hand, BrC and PCa diagnosis should not only be focused on early disease detection, but also be restrictive to clinically significant disease detection. Hence, new screening biomarkers have been exhaustively searched (Table 1, Table 2, Table 3 and Table 4). One may wonder if a minimally invasive “PanCancer” detection approach based on liquid biopsies with good sensitivity and specificity for simultaneous cancer detection would increase screening effectiveness. In fact, a study performed in the context of CRC screening revealed that only 37% of the 172 subjects were compliant with screening colonoscopy [197]. Interestingly, 97% of the subjects who refused this modality accepted a non-invasive alternative, of which 83% selected *SEPT9_me_* blood-test and only 15% the stool test, being the primary reason for this choice the convenience of the procedure [197]. Regardless of being small scaled, this study clearly demonstrates that screening compliance can dramatically increase if a convenient minimally invasive option, such as a liquid biopsy, is offered. Nevertheless, the establishment of a “PanCancer” panel for a simultaneous detection of several cancer types is a challenge given their heterogeneity.

Regarding ccfDNA methylation-based biomarkers, although only *RARβ2_me_*, and *RASSF1A_me_* levels were reported in all four cancer types (LC, BrC, CRC, and PCa), several other genes have already been reported in at least two of the four different cancer types (Figure 5).

Interestingly, a study published in 2007 that included 70 serum samples from metastatic BrC, NSCLC, gastric, pancreatic, colorectal and hepatocellular carcinoma and 10 healthy serum controls demonstrated that a gene panel hypermethylation (*RUNX3_me_, p16_me_, RASSF1A_me_ and CDH1_me_*) detected cancer samples with 89% sensitivity and 100% specificity, using MSP [198], suggesting the putative value of using a single panel to detect several malignancies.

Remarkably, genome-wide DNA methylation studies have also been performed aiming to detect the presence of cancer and underlying cancer type [199,200,201]. Indeed, Li et al. and Kang et al. developed “CancerDetector” and “CancerLocator” that may detect cancer using probabilistic approaches based on ccfDNA methylation sequencing [200,201]. Moreover, Moss et al. using Illumina methylation arrays demonstrated that plasma methylation patterns can be used to identify cell type-specific ccfDNA in healthy and pathological conditions, including different types of cancer [202].

Interestingly, in 2018, the Laboratory for Advanced Medicine launched the LTD IvyGene^®^ Test in the USA for detection of BrC, CRC, liver cancer and LC [203]. This test utilizes a multi-target approach derived from sequencing methods to detect ccfDNA methylation profile in 40mL of whole blood samples [203,204]. Notwithstanding, according to their website, the test detected cancer with 84% sensitivity and 90% specificity, it was only validated in 197 samples obtained from subjects with either no history of cancer or diagnosis of one of the four cancers [203]. Thus, the performance evaluation of IvyGene^®^ Test in a large screening setting is still warranted.

Although epigenome-wide approaches allow for simultaneous screen of several hundreds of genes and might be advantageous to boost the discovery of new differentially methylated DNA regions, they are not still widely available in clinical laboratories and require high-level bioinformatics expertise, fast data processing and large data storage capabilities [205,206]. Additionally, depending on the number of samples and genes to be analyzed, targeted approaches, such as qMSP or droplet digital MSP (ddMSP) might be more cost-effective [207]. Indeed, two recent studies aimed to develop a “PanCancer” panel to detect the most incident malignancies, evaluated methylation levels of several genes in plasma samples using multiplex qMSP. Remarkably, *APC_me_*, *FOXA1_me_* and *RASSF1A_me_* panel was able to detect BrC, CRC and LC in female patients with 72% sensitivity and 74% specificity [36], whereas *FOXA1_me_* and *RARβ2_me_* panel detected LC and PCa in males with 64% sensitivity and 70% sensitivity [46]. Although both studies included a “CancerType” panel to indicate the most likely cancer topography, reaching specificities above 80%, sensitivity to this end was rather limited [36,46].

Despite the significant advancements, standardization of technical procedures, such as amount of blood collected, type of samples to be used (plasma *vs.* serum), and DNA extraction and methylation analysis are still necessary to allow reproducibility between different studies and boost the translation of these markers into clinical practice. Nevertheless, although validation in large independent prospective cohorts by clinical trials in asymptomatic individuals to access the real clinical value of a screening method to detect several cancer types is still required, the data suggests that the hypothetical use of a “PanCancer” panel based on ccfDNA methylation is amenable to increase patient compliance to screening programs and decrease patient morbidity and mortality as well as healthcare systems costs.

## 6. Conclusions

In conclusion, these data indicate that LC, BrC, CRC, and PCa detection and patients’ stratification according to prognosis can be achieved by analyzing gene methylation levels in ccfDNA extracted from plasma or serum, constituting a minimally invasive approach. Methylation levels’ analysis in ccfDNA may not only complement current screening methods, but also aid in stratifying cancer patients according to recurrence risk and response to therapy. Nevertheless, these findings still lack validation in larger multicenter studies to enable their implementation in clinical practice.

## Figures and Tables

**Figure 1 cells-09-00624-f001:**
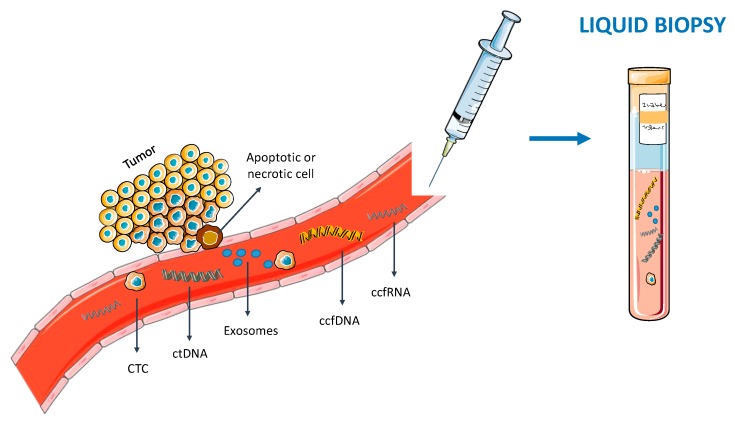
Blood-based liquid biopsy. Circulating tumor cells (CTC), circulating cell-free DNA (ccfDNA) [including circulating tumor DNA (ctDNA)], circulating cell-free RNA (ccfRNA) and exosomes are released from tumor cells to the bloodstream. Hence, blood can be collected and analyzed in the context of a liquid biopsy.

**Figure 2 cells-09-00624-f002:**
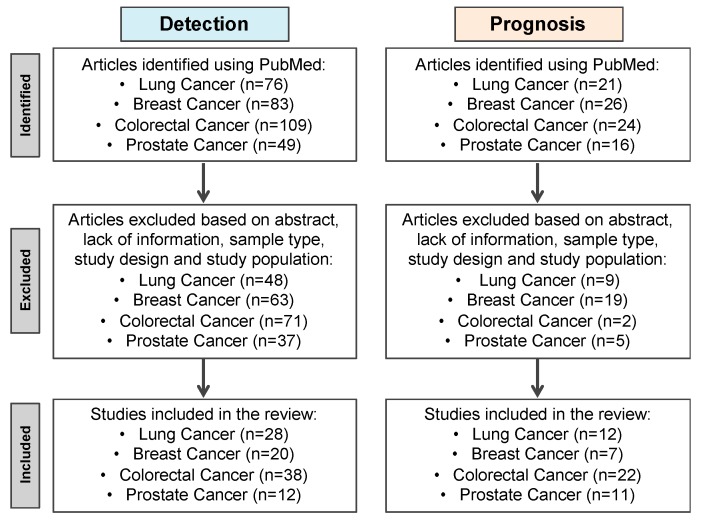
Flow diagram of Pubmed available studies’ selection procedure using the key words “Lung Cancer/Breast Cancer/Colorectal Cancer/Lung Cancer”, “DNA methylation”, “Diagnosis/Detection/Prognosis” and “Serum/Plasma”.

**Figure 3 cells-09-00624-f003:**
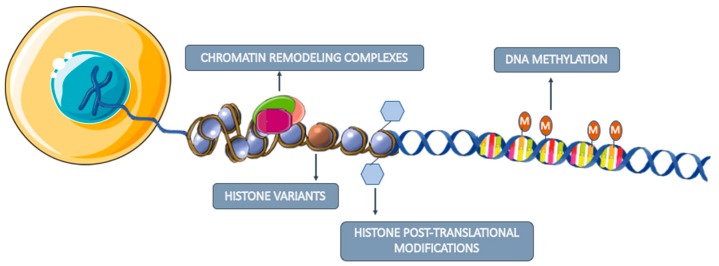
Major studied epigenetic mechanisms involved in gene expression regulation. DNA methylation consists in the addition of a methyl group to a cytosine present in a cytosine-phosphate-guanine (CpG). Histone post-translational modifications refer to the addition of biochemical modifications on histone tails, such as methylation, acetylation, phosphorylation, ubiquitylation and SUMOylation that regulate gene expression. Histone variants differ a few amino acids from canonical histones and regulate chromatin remodeling and histone post-translational modifications. Chromatin remodeling complexes regulate the nucleosome structure by removing, relocate and shifting histones.

**Figure 4 cells-09-00624-f004:**
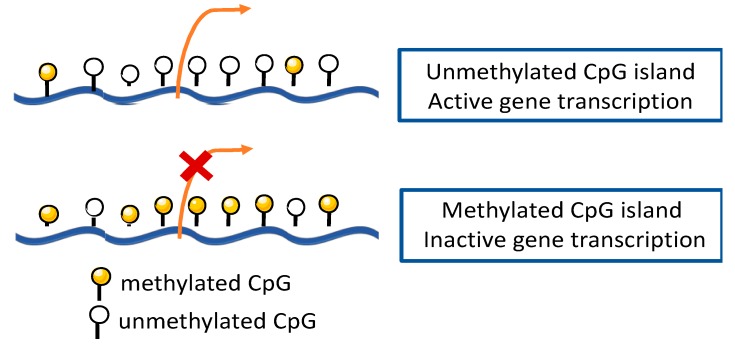
DNA methylation within a gene promoter region. Unmethylated CpG islands enable gene transcription. When CpG island is methylated, gene transcription is repressed.

**Figure 5 cells-09-00624-f005:**
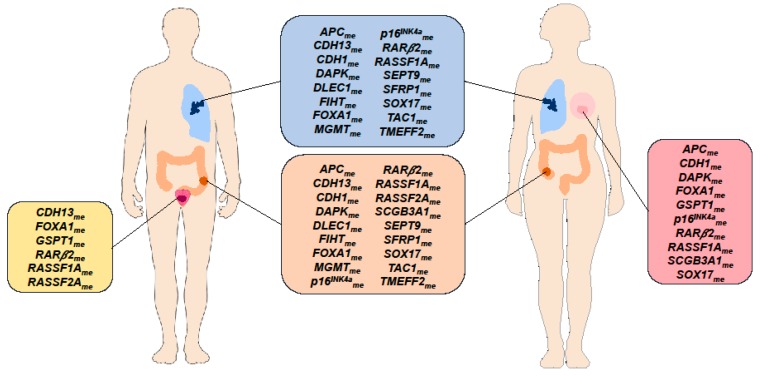
Circulating cell-free DNA methylation-based biomarkers described in the literature for cancer detection common to at least two cancer types [Breast Cancer (pink box), Lung Cancer (blue box), Prostate Cancer (yellow box), Colorectal Cancer (orange box)].

**Table 1 cells-09-00624-t001:** CcfDNA-based methylation biomarkers for lung cancer (LC) detection.

Lung Cancer						
Genes	Number of Cases/Controls	Sensitivity (%)	Specificity (%)	Sources	Methods	References
*APC_me_*	89 LC/50 AC	47	100	Serum/Plasma	qMSP	[33]
*p16^INK4a^_me_*	35 NSCLC/15 AC	34	100	Plasma	F-MSP	[34]
*MGMT_me_/p16^INK4a^_me_/RASSF1A_me_/DAPK_me_/RARβ_me_*	91 LC/109 BPD	50	85	Serum	MSP	[47]
*p16^INK4a^_me_/CDH13_me_*	61 NSCLC/15 BPD	39	100	Serum	MSP	[48]
*RASSF1A_me_*	80 LC/50 AC ^a^	34	100	Serum	MSP	[49]
*CDH13_me_/p16^INK4a^_me_/FHIT_me_/* *RARβ_me_* */RASSF1A_me_/ZMYND10_me_*	63 NSCLC/36 BPD	73	83	Plasma	Two-step MSP	[50]
*KLK10_me_*	78 NSCLC/50 AC ^a^	38	96	Plasma	MSP	[51]
*SFRP1_me_*	78 NSCLC/50 AC ^a^	28	96	Plasma	MSP	[52]
*DLEC1_me_*	78 NSCLC/50 AC ^a^	36	96	Plasma	MSP	[53]
*Kif1a_me_/DCC_me_/RARβ2_me_/NISCH_me_*	70 LC/80 BPD	73	71	Plasma	qMSP	[54]
*APC_me_/RASSF1A_me_/CDH13_me_/* *KLK10_me_/DLEC1_me_*	110 NSCLC ^b^/50 AC ^a^	84	74	Plasma	MSP	[55]
*APC_me_/CDH1_me_/MGMT_me_/DCC_me_ RASSF1A_me_/AIM1_me_*	76 LC/30 AC	84	57	Serum	qMSP	[56]
*SHOX2_me_*	188 LC/155 AC ^a,c^	60	90	Plasma	qMSP	[37]
*TMEFF2_me_*	316 NSCLC/50 AC	9	100	Serum	Two-step MSP	[57]
*RARβ2_me_*	60 NSCLC/32 AC	72	62	Plasma	qMSP	[35]
*RASSF1A_me_*		66	57			
*SEPT9_me_*	70 LC/100 AC	44	92	Plasma	qMSP	[43]
*p14ARF_me_*	107 NSCLC/20 BPD	25	95	Plasma	Two-step MSP	[58]
*DCLK1_me_*	65 LC/95 AC	49	92	Plasma	qMSP	[42]
*SOX17_me_*	48 Operable NSCLC/49 AC	56	98	Plasma	qMSP	[59]
	74 Advanced NSCLC/49 AC	36				
*SHOX2_me_*	38 LC/31 BPD	81	79	Plasma	qMSP	[38]
*SHOX2_me_/PTGER4_me_*	50 LC/122 AC ^a^	67 90	90 73	Plasma	Multiplex qMSP	[39]
*CDO1_me_/TAC1_me_/SOX17_me_*	150 NSCLC ^b^/60 AC	93	62	Plasma	qMSP	[60]
*MARCH11_me_/HOXA9_me_/CDO1_me_/* *UNCX_me_/PTGDR_me_/AJAP1_me_*	43 LUAD ^d^/42 AC	72	71	Plasma	qMSP	[44]
	40 LUSC ^d^/42 AC	60				
*NID2_me_*	46 NSCLC/30 BPD	46	80	Plasma	qMSP	[61]
*APC_me_*	73 LC ^e^/103 AC ^e^	36	94	Plasma	Multiplex qMSP	[36]
*FOXA1_me_*		72	74			
*RARβ2_me_*		25	95			
*RASSF1A_me_*		22	98			
*SOX17_me_*		38	95			
*CDH13_me_/WT1_me_/CDKN2A_me_/HOXA9_me_/* *PITX2_me_/CALCA_me_/RASSF1A_me_/DLEC1_me_*	39 LC/11 BPD	72	91	Plasma	qMSP	[62]
*APC_me_/RASSF1A_me_*	129 LC/28 BDP	38	93	Plasma	qMSP	[45]
*FOXA1_me_/RARβ2_me_/RASSF1A_me_/SOX17_me_*	102 LC ^f^/136 AC ^f^	66	70	Plasma	qMSP	[46]

^a^ Included Benign pulmonary diseases; ^b^ Only stage I/II; ^c^ Included other cancer types; ^d^ Only included stage I; ^e^ Only included females; ^f^ Only included males; Abbreviations: AC—Asymptomatic Controls; BPD—Benign Pulmonary Diseases; F-MSP—Fluorescent methylation-specific PCR; LC—Lung Cancer; LUAD—Lung Adenocarcinoma; LUSC—Lung Squamous Cell Carcinoma; MSP—Methylation-specific PCR; NSCLC—Non-Small Cell Lung Cancer; qMSP—Quantitative methylation-specific PCR.

**Table 2 cells-09-00624-t002:** CcfDNA-based methylation biomarkers for breast cancer (BrC) detection.

Breast Cancer						
Genes	Number of Cases/Controls	Sensitivity (%)	Specificity (%)	Sources	Methods	References
*APC_me_/DAPkinase_me_/RASSF1A_me_*	34 BrC/20 AC + 8 Benign	94	100	Serum	MSP	[83]
*ATM_me_/RASSF1A_me_*	50 BrC/14 AC	36	100	Plasma	qMSP	[84]
*RARβ2_me_/RASSF1A_me_*	20 BrC/10 AC	95	100	Plasma ^a^	MSP	[85]
*APC_me_/GSPT1_me_/RARβ2_me_*/*RASSF1A_me_*	47 BrC/38 AC	62	87	Plasma	qMSP	[77]
*14-3-3-σ_me_/ESR1_me_^b^*	106 BrC/74 AC	81	55	Serum	qMSP	[80]
*APC_me_/ESR1_me_/RASSF1A_me_*	79 BrC/19 AC	53	84	Serum	qMSP	[86]
*RASSF1A_me_*	61 BrC/29 AC	18	100	Plasma	MSP	[88]
*DAPK1_me_/RASSF1A_me_*	26 BrC/16 AC 26 BrC/12 Benign	96	92	Serum	MSP	[81]
			57			
*APC_me_/BIN1_me_/BRCA1_me_/CST6_me_/GSTP1_me_/p16 (CDKN2A)_me_/p21 (CDKN1A)_me_/TIMP3_me_*	36 BrC/30 AC	91.7	-	Plasma	Mass Spectrometry	[89]
*RARβ_me_/RASSF1A_me_*	119 BrC/125 AC	94.1	88.8	Serum	Two-step qMSP	[78]
*GSTP1_me_/RARβ2_me_/RASSF1A_me_*	101 BrC ^c^/87 AC	22	93	Serum	One-step MSP	[90]
*SOX17_me_*	114 BrC/49 AC	38	98	Plasma	qMSP	[91]
*ITIH5_me_/DKK3_me_/RASSF1A_me_*	138 BrC/135 AC	67	69	Serum	qMSP	[79]
	138 BrC/39 Benign		82			
*APC_me_* *RARβ2_me_*	121 BrC/66 AC + 79 Benign	93.4	95.4	Serum	MSP	[92]
		95.6	92.4			
*SFN_me_/p16_me_/hMLH1_me_/HOXD13_me_/PCDHGB7_me_/RASSF1A_me_*	125 BrC/104 Benign	82.4	78.1	Serum	qMSP	[87]
	125 BrC/104 AC	79.6	72.4			
*CDH1_me_/RASSF1A_me_*	50 BrC/25 AC	76	90	Serum	MSP	[93]
*NBPF1_me_*	52 BrC ^c^/30 AC	67.1 ^d^	59.1 ^d^	Plasma	MSP	[94]
*APC_me_*	108 BrC/103 AC	32.4	94.2	Plasma	Multiplex qMSP	[36]
*FOXA1_me_*		38.9	79.6			
*RASSF1A_me_*		19.4	100			
*SCGB3A1_me_*		21.3	92.2			
*APC_me_/FOXA1_me_/RASSF1A_me_*	44 BrC/39 AC	81.8	76.9	Plasma	Multiplex qMSP	[82]
*PER1_me_/NKX2-6_me_/SPAG6_me_*	111 BrC/14 Benign	58	79	Plasma	Pyrosequencing	[95]

^a^ Total circulating DNA, including cell-bound circulating DNA; ^b^ Considering *ER-α_me_* or *14-3-3-σ_me_*; ^c^ Stages I-III; ^d^ For one gene site. Abbreviations: AC—Asymptomatic Controls; BrC—Breast cancer; MSP—Methylation-specific PCR; qMSP—Quantitative methylation-specific PCR.

**Table 3 cells-09-00624-t003:** CcfDNA-based methylation biomarkers for colorectal cancer (CRC) detection.

Colorectal Cancer						
Genes	Number of Cases/Controls	Sensitivity (%)	Specificity (%)	Sources	Methods	References
*p16^INK4a^_me_*	52 CRC/44 AC ^a^	27	100	Serum	MSP	[122]
*APC_me_/hMLH1_me_/HLTF_me_*	49 CRC/41 AC	57	90	Serum	qMSP	[123]
*ALX4_me_*	30 CRC/30 AC	83	70	Serum	qMSP	[124]
*HPP1_me_*	38 CRC/20 AC	49	100	Serum	qMSP	[125]
*HLTF_me_*		67				
*hMLH1_me_*		47				
*SEPT9_me_*	133 CRC/179 AC	69	86	Plasma	qMSP	[126]
*TMEFF2_me_*		65	69			
*NGFR_me_*		51	84			
*RASSF1A_me_*	45 CRC/30 AC	29	100	Serum	MSP	[127]
*VIM_me_*	81 CRC/110 AC	59	93	Plasma	Methyl BEAMing	[128]
*APC_me_/MGMT_me_/RASSF2A_me_/* *WIF1_me_*	243 CRC ^b^/276 AC	87	92	Plasma	MSP	[118]
	64 Adenoma/276 AC	75				
*ALX4_me_/SEPT9_me_/TMEFF2_me_*	182 CRC/170 AC	81	90	Plasma	Multiplex qMSP	[129]
*NEUROG1_me_*	97 CRC ^b^/45 AC	61	91	Serum	qMSP	[130]
*TFPI2_me_*	215 CRC/20 AC	18	100	Serum	qMSP	[131]
*DLC1_me_*	85 CRC/45 AC	42	91	Serum	MSP	[132]
*CYCD2_me_/HIC1_me_/PAX5_me_/* *RASSF1A_me_/RB1_me_/SRBC_me_*	30 CRC ^b^/30 AC	84	68	Plasma	Microarray	[133]
*HIC1_me_/MDG1_me_/RASSF1A_me_*	30 Adenoma/30 AC	55	65			
*SMAD4_me_*	60 CRC/100 AC ^a^	52	64	Plasma	MSP-SSCP	[134]
*FHIT_me_*		50	84			
*DAPK1_me_*		50	74			
*APC_me_*		57	86			
*CDH1_me_*		60	84			
*SDC2_me_*	131 CRC/125 AC	87	95	Serum	qMSP	[135]
*TAC1_me_/SEPT9_me_*	26 CRC ^c^/26 AC	73	92	Serum	qMSP	[136]
*NPY_me_/PENK_me_/WIF1_me_*	32 CRC/161 AC	87 59	80 95	Serum	Multiplex qMSP	[137]
*CAHM_me_*	73 CRC/74 AC	55	93	Plasma	qMSP	[138]
	73 Adenoma/74 AC	4				
*PPP1R3C_me_/EFHD1_me_*	120 CRC/96 AC	53 (2 genes)	96 (2 genes)	Plasma	MSP	[139]
		90 (at least 1 gene)	64 (at least 1 gene)			
*SYNE1_me_/FOXE1_me_*	66 CRC/140 AC	58	91	Plasma	Multiplex qMSP	[140]
*GATA5_me_/SFRP2_me_*	57 CRC/47 AC	43	91	Plasma	MSP	[141]
	30 Adenoma/47 AC	27				
*BCAT1_me_/IKZF1_me_*	74 CRC/144 AC	77	92	Plasma	qMSP	[142]
*BCAT1_me_/IKZF1_me_*	129 CRC/450 AC	66	95	Plasma	qMSP	[120]
	338 Advanced Adenoma/450 AC	6				
	346 Non-Advanced Adenoma/450 AC	7				
*BCAT1_me_/IKZF1_me_*	66 CRC/1315 AC ^a^	62	92	Plasma	qMSP	[121]
	170 Advanced Adenoma	9	---			
	278 Non-Advanced Adenoma	9	---			
*ALX4_me_*	25 CRC/25 AC	68	88	Serum	MSP	[143]
*FGF5_me_*	20 CRC/40 AC	85	82	Plasma	qMSP	[144]
*GRASP_me_*	44 CRC/44 AC	55	93	Plasma	qMSP	[145]
*IRF4_me_*	22 CRC/24 AC	59	96			
*PDX1_me_*	20 CRC/20 AC	45	70			
*SDC2_me_*	44 CRC/44 AC	59	84			
*SEPT9_me_*	44 CRC/44 AC	59	95			
*SOX21_me_*	20 CRC/20 AC	85	50			
*SPG20_me_*	37 CRC/37 AC	81	97			
*SEPT9_me_*	Meta-analysis	78 (1/3)	84 (1/3)	Plasma/ Serum	---	[113]
		73 (2/3)	96 (2/3)			
*ALX4_me_/BMP3_me_/NPTX2_me_/RARβ_me_/SDC2_me_/SEPT9_me_/VIM_me_*	193 CRC/102 AC	91	73	Plasma	Two-step qMSP	[146]
*SFRP1_me_/SFRP2_me_/SDC2_me_/* *PRIMA1_me_*	47 CRC/37 AC	92	97	Plasma	qMSP	[119]
	37 Adenoma/37 AC	89	87			
*BMP3_me_*	45 CRC/50 AC	40	94	Plasma	BS-HRM	[147]
*TWIST1_me_*	18 CRC/25 AC	44	92	Serum	Multiplex ddMSP	[117]
	70 Advanced Adenoma/25 AC	30				
	25 Non-Advanced Adenoma/25 AC	36				
*SEPT9_me_*	98 CRC/253 AC	61	98	Plasma	qMSP	[117]
	101 Adenoma/253 AC	8				
*APC_me_*	72 CRC ^d^/103 AC ^d^	21	94	Plasma	Multiplex qMSP	[36]
*FOXA1_me_*		50	88			
*RARβ2_me_*		17	95			
*RASSF1A_me_*		14	99			
*SCGB3A1_me_*		26	90			
*SEPT9_me_*		11	100			
*SOX17_me_*		24	90			
*SFRP2_me_*	62 CRC/55 AC	69	87	Serum	qMSP	[148]
*SEPT9 _me_*/*SDC2_me_*	117 CRC/166 AC	88.9	92.9	Plasma	qMSP	[149]
*C9orf50_me_/KCNQ5_me_/CLIP4_me_*	143 CRC/91 AC	91	99	Plasma	ddMSP	[150]
*SEPT9_me_/SOX17_me_*	100 CRC ^e^/136 AC ^e^	12	100	Plasma	Multiplex qMSP	[46]

^a^ Included patients with adenomatous polyps; ^b^ Only included stages I/II; ^c^ Only included stage I; ^d^ Only included females; ^e^ Only included males; Abbreviations: AC – Asymptomatic Controls; Adenoma – Adenomatous polyps; BS-HRM – Bisulfite specific high-resolution melting analysis; CRC – Colorectal Cancer; ddMSP – Digital droplet methylation-specific PCR; MSP – Methylation-specific PCR; MSP-SSCP – Methylation-Specific PCR – single strand conformation polymorphism; qMSP – Quantitative methylation-specific PCR.

**Table 4 cells-09-00624-t004:** CcfDNA-based methylation biomarkers for prostate cancer (PCa) detection.

Prostate Cancer						
Genes	Number of Cases/Controls	Sensitivity (%)	Specificity (%)	Sources	Methods	References
*GSTP1_me_/PTGS2_me_/RPRM_me_/TIG1_me_*	168 PCa/42 BPH	47	93	Serum	qMSP	[172]
*MDR1_me_*	192 PCa/35 AC ^a^	32	100	Serum	qMSP	[181]
*GSTP1_me_/RASSF1A_me_/RARβ2_me_*	83 PCa/40 AC	29	100	Serum	MSP	[173]
*GSTP1_me_*	80 PCa/51 AC ^a^	26	80	Plasma	qMSP	[175]
*RASSF2A_me_*		28				
*HIST1H4K_me_*		17				
*TFAP2E_me_*		12				
*GSTP1_me_*	Meta-analysis	40	90	Plasma/Serum	Non-qMSP	[174]
		36	96		qMSP	
*RARβ2_m_*	91 PCa/94 BPH	93	89	Serum	qMSP	[182]
*GSTP1_me_*	31 PCa/44 BPH	93	89	Plasma	MSP	[176]
*CDH13_me_*	98 PCa/47 AC ^b^	45	100	Serum	MSP	[183]
*GADD45a_me_*	34 PCa/48 BPH	38	98	Serum	Pyrosequencing	[180]
*MCAM_me_/ERα_me_/ERβ_me_*	84 PCa/30 AC	75	70	Serum	qMSP	[177]
*CCDC181_me_/ST6GALNAC3_me_/HAPLN3_me_*	27 PCa/10 BPH	67	100	Serum	ddMSP	[178]
*ZNF660_me_*		22				
*FOXA1_me_/RARβ2_m_/RASSF1A_me_/GSTP1_me_*	121 PCa/136 AC	72	72	Plasma	Multiplex qMSP	[46]

^a^ Biopsy negative; ^b^ Included BPH; Abbreviations: AC—Asymptomatic Control; BPH—Benign Prostatic Hyperplasia; ddMSP—Digital droplet methylation-specific PCR; MSP—Methylation-specific PCR; PCa—Prostate Cancer; qMSP—Quantitative methylation-specific PCR.

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
