# Peer review of "DNA Methylation-Based Testing in Liquid Biopsies as Detection and Prognostic Biomarkers for the Four Major Cancer Types"

_cells, 2020, doi:10.3390/cells9030624_

Round 1
Reviewer 1 Report
The manuscript entitled "DNA methylation-based testing in liquid biopsies as detection and prognostic biomarkers for the four major cancers types" by Constancio et al inherited about a systemativ revision of literature data to evaluate DNA methylation-based biomarkers promising role for cancer detection and management is well written and suitable for pubblication after minor revisions: - for each tumor, could the authors better define the real clinical application respect to the promising biomarker evaluation ? In my opinion a clarification about the clinical practice yet apporved approach is necessary for the focus of the manuscript - In the discussion setting , please, could the authors report if other relevant studies were approached to evaluate methylation profile starting from other analytes in liquid biopsy ? ( exosomes, platelet derived cf DNA...) - In the manuscript could the authors verify if "Error! Reference source not found"?Author Response
Please see the attachment

Reviewer 2 Report
This is a comprehensive review regarding the biomarker potential of cell free circulating DNA methylation in four major types of cancer. The manuscript is well written; however, since it addresses four types of cancer, each being a highly heterogeneous diseases with various subtypes, the paper is over 40 pages long without references. Perhaps too long for a journal and more suitable as a book chapter. If they have focused on only one or two of these cancers, then a more in-depth discussion of the various biomarkers for each cancer molecular subtype and stage could have been systematically presented in tabular format more easy to follow, containing more relevant information such as positive predictive value, AUC values, and other biomarker characteristics.
A schematic representation of their search strategy or at least description of the results obtained through their pubmed search (how many papers they identified, how many excluded, why, etc.) should be included.
Only minor language issues were detected:
Title: Use singular for cancer in the title, “…major cancer types”
Line 55, there seems to be an error with the references, which also appear later on, so please double check references throughout.
Line 71, the correct term is “state of the art”
Line 72, change prognostication to prognosis
Line 73, change monitorization to monitoring
Line 117-118, only the legend of Figure 3 is shown, and then the figure 3 is shown later on in line 125-126 and the legend is repeated.
Tables containing information regarding to Prognosis, Prediction and Monitoring would be a welcome addition for the reader.
